# Assessing Entomological and Epidemiological Efficacy of Pyriproxyfen-Treated Ovitraps in the Reduction of *Aedes* Species: A Quasi-Experiment on Dengue Infection Using Saliva Samples

**DOI:** 10.3390/ijerph19053026

**Published:** 2022-03-04

**Authors:** Antonio D. Ligsay, Kristan Jela M. Tambio, Michelle Joyce M. Aytona, Grecebio Jonathan D. Alejandro, Zypher Jude G. Regencia, Emmanuel S. Baja, Richard Edward L. Paul

**Affiliations:** 1The Graduate School & College of Science, University of Santo Tomas España Boulevard, Manila 1008, Philippines; gdalejandro@ust.edu.ph; 2Clinical Research Section, St. Luke’s College of Medicine—William H. Quasha Memorial, 279 E. Rodriguez Sr. Ave, Quezon City 1112, Philippines; kristan.tambio@gmail.com (K.J.M.T.); mjaytona@gmail.com (M.J.M.A.); 3Department of Clinical Epidemiology, College of Medicine, University of the Philippines Manila, Pedro Gil Street, Ermita, Manila 1000, Philippines; zgregencia@up.edu.ph (Z.J.G.R.); esbaja@up.edu.ph (E.S.B.); 4Institute of Clinical Epidemiology, National Institutes of Health, University of the Philippines Manila, 623 Pedro Gil Street, Ermita, Manila 1000, Philippines; 5Institut Pasteur, Functional Genetics of Infectious Diseases Unit, UMR 2000 (CNRS), 75015 Paris, France; richard.paul@pasteur.fr

**Keywords:** mosquito control, dengue, mosquito density, pyriproxyfen, Philippines

## Abstract

Our study assessed the impact of using ovitraps with pyriproxyfen on mosquito populations and the feasibility of using human saliva samples to test for seroconversion to dengue virus (DENV). We used a quasi-experimental research design by forming the intervention (*n* = 220) and the control (*n* = 223) groups in neighboring Taguig City, Philippines, over 4 months. Socio-demographic data, entomological indices, and IgG antibodies against DENV were measured. Associations between the implementation of ovitraps dosed with pyriproxyfen and mosquito densities (percentage positive ovitraps and container indices) and DENV seroconversion were calculated post-intervention in Months 2, 3, and 4. Among the participants recruited at baseline, 17 and 13 were seropositive for dengue (DENV) in the intervention and control groups, respectively. Both entomological indices were lower in the treated area than the control site at post-intervention Months 2, 3, and 4, but not earlier. Dengue seroconversions rates decreased in the treated population, but not significantly so. In conclusion, the use of PPF-treated ovitraps may have impacted the mosquito population, but not seroconversion rates. Compliance in providing saliva samples and the ability to detect IgG antibodies within these samples was encouraging and suggests that further studies on larger populations for longer durations are warranted.

## 1. Introduction

Over the last five decades, the incidence of dengue has increased more than 30-fold, causing an estimated 50–100 million cases annually that are spread across more than 100 endemic countries [1]. This substantial increase in the burden of dengue is considered to have been exacerbated by increased international transport and travel that has facilitated the global spread of the mosquito vector spp. and the virus [2]. Moreover, because the tropical vector, *Aedes aegypti*, has adapted well to urban environments, increasing levels of urbanization expose an ever-increasing population to the dengue virus (DENV) [3,4,5]. The impact of urban mosquito-borne diseases upon public health is not only restricted to dengue, as evidenced by the recent outbreaks of other arboviruses, notably Zika and chikungunya, along with the continuing burden of urban malaria in India by *Anopheles stephensi*, which has also adapted to urban environments [6,7,8,9,10,11]. The threat from arboviruses in urban settings is no longer restricted to the tropics and sub-tropics, as many invasive mosquito vectors can survive the temperate climates found in their newly invaded habitat [12,13,14]. Outbreaks of many arboviral diseases, including dengue, chikungunya, and West Nile, have occurred in Europe and the US [15,16,17]. Additionally, global climate warming is projected to expand the range of mosquito vectors into new areas extending the transmission season in current endemic settings, and increasing the mosquito’s vectorial capacity [18,19]. Thus, urban mosquito-borne diseases are projected to impose an ever-increasing burden upon public health within affected societies and require urgent attention [14]. With the absence of buy-in for potentially effective vaccines, reducing human contact with vectors is the only way to reduce the burden of dengue and other mosquito-borne diseases [20].

Current mosquito control approaches include the use of larvicides in stored water containers to eliminate juvenile-stage mosquitoes, adulticides to kill adult-stage mosquitos, and source reduction via improved environmental hygiene (eliminating the solid waste that offers potential aquatic habitats for larvae). Adulticides are generally employed in and around houses of clinical cases identified through the public health system and provide short-term control. In contrast, source reduction through eliminating oviposition sites before the rainy season has shown some success but requires enormous effort [21]. Insect growth regulator mimics, such as novularon, methoprene, and pyriproxyfen (PPF) have been used extensively in mosquito control programs worldwide [22,23,24,25,26]. Granular or liquid formulations of PPF applied to water containers and drains have had documented success in reducing immature and adult mosquito numbers and have been associated with a reduction in dengue incidence [22].

In response to the growing global threat imposed by mosquito-borne diseases and recognizing that the current methods are inadequate, the World Health Organization launched its Global Vector Control Response 2017–2030 to promote the development of novel approaches that show not only entomological but also epidemiological efficacy [23].

Successful measurement of epidemiological efficacy is challenging. A substantial but variable proportion of DENV infections are subclinical. Thus, the sample size required to detect the effect of mosquito control by measuring disease incidence would need to be very large. Seroconversion rates provide a more accurate measure, but the spatial and temporal unpredictable heterogeneity in dengue epidemiology means that repeated sampling from individuals is required to generate an interpretable result. Repeated blood sampling is problematic because of compliance. Recently, saliva samples have been proposed to detect antibodies against DENV and offer a non-invasive alternative to blood sampling [24].

Here we present a study in Taguig City, Philippines, to assess and demonstrate the potential epidemiological efficacy of this approach and to determine the period of time necessary to observe any entomological impact incurred through the use of ovitraps laced with PPF.

## 2. Materials and Methods

### 2.1. Study Design, Sites, and Population

Our study used a quasi-experimental research design composed of two groups, with and without intervention. Data on dengue incidence were obtained from the Department of Health (DOH)—Metro Manila Centers for Health Development. Cities in the National Capital Region with the highest dengue attack rate were candidate areas for the study. Taguig City, a densely populated area in the south of Manila, was chosen. Criteria for selecting sites within Taguig included small municipalities/barangays which constantly reported 10–50 dengue cases on a monthly basis. The barangay of Western Bicutan was chosen; it contains approximately 100,000 residents. The Katipunan area was selected as the control site and the Philippines National Railways area as the intervention site. These sites are in close proximity to one another but are not juxtaposed. Intervention and control sites were matched as closely as possible according to the history of local dengue incidence. Once a dengue index case was documented within a site, ovitraps (mosquito density monitoring devices) were installed for two weeks to assess whether mosquito densities (>5% ovitraps positive for mosquito larvae) were sufficiently high for active DENV transmission and therefore selectable as a study site. Figure 1a,b show the schematics of the two study sites with a distribution of ovitraps to measure suitability for study once a dengue index case was detected.

### 2.2. Inclusion and Exclusion Criteria of Participants

#### 2.2.1. Inclusion Criteria

After selecting the site, we invited participants living within 100 m of the dengue index case. Our target study population was 200 per intervention site. Invited participants were individuals aged 1–30 years old who were willing to participate in the study and could provide saliva samples for DENV antibody detection. This age group was chosen to maximize the probability of identifying individuals never having had a previous DENV infection. A baseline saliva sample was taken at the time of recruitment and kept on ice before storage at −80 °C while awaiting analysis.

#### 2.2.2. Exclusion Criteria

At the time of collection, individuals who were febrile, suffering from any chronic disease, or refused to sign the informed consent form were excluded.

### 2.3. Data Collection

#### 2.3.1. Entomological Indices, Intervention, and Control Ovitap Set-Up

Following the parameters set by the WHO, different entomological indices were measured. We used the container index and the ovitrap index. The container index is designed to measure mosquito larval abundance and characterize the typology of larval development habitats. To measure the container index, all identifiable water-filled containers are inspected for the presence of mosquito larvae and the index is the number of containers infested/total containers inspected and given as a percentage. Ovitraps are economical and sensitive devices that enable surveillance of the spread and seasonal dynamics of *Aedes aegypti*. The formula to measure the ovitrap index is the number of ovitraps positive for mosquito eggs/the total number of ovitraps examined in a given area per unit of time (month/week), given as a percentage [25]. Ovi-PPF devices (ovitraps filled with 250 mL of water with one sachet of Sumilarv™ 0.5 G Sumitomo Chemical) were placed in the selected households in the intervention study site houses at the time of recruitment. In the control site, ovitraps without PPF solution were similarly deployed. One trap was placed per house indoors. Ovitraps were inspected weekly by barangay health workers, and the contents were serviced every two weeks. Should the contents be less than the required volume (half full), the ovitrap (with or without PPF) would be refilled. Container indices were calculated on a weekly basis by inspecting the number of water-filled containers positive for mosquito larvae/pupae. Water-field containers found positive for larvae were then emptied and were, if used for water storage, then re-filled.

Participating houses that were directly adjacent were combined into clusters and mosquito indices were calculated at a cluster level. If there were no containers containing water in a cluster at a given time, the CI was marked as missing data rather than zero. Weekly cluster level indices were averaged over the given month. Data collection was done during the cool-dry season, from June to September 2019.

#### 2.3.2. Outcome Assessment: Antibody Measurements

A saliva sample for baseline IgG determination was carried out among participating individuals in the intervention and control groups at Month 0. At that time, information on the participants’ age, gender, household income, and their parents’ levels of education were ascertained. Individuals were anonymized with assigned codes. Ovitraps with PPF were then deployed. The first post-intervention saliva monitoring was done after two months and then monthly for a total of three monitoring visits. Thus, the Ovi-PPF traps were in place for a total of four months.

The IgG ELISA test was done according to the procedures previously described [24]. Positive and negative controls previously confirmed as control references were provided by Institut Pasteur Cambodia [26]. Samples were analyzed in duplicates. In case there were conflicting results, a third test was performed. An optical density (OD) difference between the sample and the control of 0.1 and above was considered positive for IgG. Only those participants who were IgG negative during the baseline sample were included in the subsequent analysis. Likewise, individuals seroconverting later (Months 2 and 3) were removed from further sampling and analysis.

### 2.4. Data Analysis

Descriptive statistics for the baseline study characteristics of the participants were calculated. In addition, ovitrap and container indices were first tested for normality using the Shapiro–Wilk test. As they were not normally distributed, these mosquito indices were arcsine transformed. The association of site type (control vs. intervention) with mosquito indices was analyzed by fitting a generalized linear model with a normal distribution, identity link function, and the significance was assessed through F statistics.

For the baseline seropositive analysis, bivariate analyses were performed. Moreover, the following explanatory variables were considered: age (child, 1–12 years old; adolescent, 13–18 years old; young adult, 19–30), sex (male; female), father’s education (no education to high school graduate; college to graduate school), mother’s education (no education to high school graduate; college to graduate school), family income (no income to <PHP 15,000.00; PHP 15,000.00 and above [1 USD = 52.00]), and site type. generalized linear models were fitted to estimate the effects of every explanatory variable on seropositivity at the baseline.

Seroconversion data 2, 3, and 4 months post-intervention (or control) were analyzed by fitting a generalized linear mixed model (GLMM) with a binomial distribution and logit function. Moreover, bivariate and multivariable analyses were performed. In the multivariable model, the following explanatory variables were considered: age (child, 1–12 years old; adolescent, 13–18 years old; young adult, 19–30), sex (male; female), father’s education (no education to high school graduate; college to graduate school), mother’s education (no education to high school graduate; college to graduate school), family income (no income to <PHP 15,000.00; PHP 15,000.00 and above), site type, month, the arcsine transformed mosquito indices, the interactions between site types, and months. House number was fitted as a random term.

In addition, IgG optical density, not being normally distributed, was analyzed by Poisson regression with an individual ID fitted as the random term in the GLMM [27]. In all cases, a dispersion parameter was estimated to account for any over-dispersion in the data. Finally, overall significance was assessed by calculating Wald statistics, and stratum level differences (i.e., between levels within a factor) were calculated through t-tests. Analyses were performed using Genstat Version 20 software (VSN International Limited; Hemel Hempstead, UK).

### 2.5. Ethical Considerations

Ethical approval was sought from an independent research ethics committee. The study was conducted in compliance with the procedures outlined in this protocol and the National Ethical Guidelines for Health Research 2011 following the Data Privacy Act of 2012.

## 3. Results

The study recruited 443 participants, of which 220 belonged to the intervention group and 223 belonged to the control group. These individuals were spatially distributed into 41 control area clusters and 48 intervention area clusters, where a cluster is defined as having participating individuals in directly-adjacent houses. The number of houses per cluster ranged from 1 to 10 with a median of 4 among a total of 327 participating houses. The number of participating individuals per house ranged from one to five, with 76.5% of houses having a single participating individual. As the study progressed and participants who seroconverted were subsequently excluded from future analyses, the total number of participants and houses decreased, reaching 281 houses located in 41 control and 46 intervention clusters, with a total of 357 individuals by the final month of the study.

At the baseline, 17 and 13 individuals were IgG positive in the intervention and control groups, respectively. These seropositive individuals were distributed across 11 houses in 7 clusters in the control site and 17 houses in 10 clusters in the intervention site. Of the 11 control site houses with a seropositive individual, five houses had more than one person tested, of which none had more than one seropositive individual. Of the 17 positive intervention site houses, seven had more than one tested individual, and only one house had more than one person seropositive.

Table 1 gives the baseline characteristics of the participants. The intervention and control groups had an average age (±SD) of 14.6 (±7.1) years old and 13.9 (±8.4) years old, respectively (*p*-value = 0.39). Females comprised a similar and higher number of participants in both the intervention (53%) and control (54%) groups (*p*-value = 0.89). The intervention group’s average monthly household income (±SD) was Philippine Pesos (PHP) 12,773 ± 13,465, compared to PhP 12,070 ± 7467 in the control group. Additionally, the categorization of household incomes revealed that most of the participants in both the intervention (73%) and control group (55%) fell under the PHP 10,001 to <20,000 income bracket. Further analysis revealed that, overall, the average monthly household income for the participants differed significantly between intervention and control groups (*p*-value < 0.01) (Table 1). The average baseline IgG ODs for each of the groups were not significantly different (*p*-value = 0.55): 0.046 (±0.036) g/L and 0.044 (±0.033) g/L for the intervention and control groups, respectively.

In the univariate statistical analyses, there was no association of seropositivity at the baseline with regards to the categories of sex (χ^2^_1_ = 0.08, *p*-value = 0.371), age (χ^2^_2_ = 0.25, *p*-value = 0.881), income (χ^2^_2_ = 2.33, *p*-value = 0.313), mother’s education (χ^2^_1_ = 0.08, *p*-value = 0.773), father’s education (χ^2^_1_ = 0.01, *p*-value = 0.941), or site type (χ^2^_1_ = 0.52, *p*-value = 0.571). No multivariable analyses were subsequently carried out as no variables had a *p*-value of < 0.25 in the univariate analysis.

As shown in Figure 2, the container indices were not significantly different in control and intervention areas at the baseline (F_1,44_ = 1.43, *p*-value = 0.239), but fractionally higher in the intervention area 1-month post-intervention (F_1,56_ = 4.18, *p*-value = 0.046). In the control areas, the CIs increased over time. By contrast, they decreased in the intervention areas and were significantly lower by the first post-intervention sampling 2 months later (F_1,49_ = 11.64, *p*-value = 0.001), and increasingly so in Months 3 (F_1,45_ = 24.15, *p*-value < 0.001) and 4 (F_1,42_ = 26.6, *p*-value < 0.001).

As shown in Figure 3, the baseline percentages of positive ovitraps was not different between the control and intervention areas (F_1,89_ = 1.92, *p*-value = 0.17), but started to become higher in the control areas in Month 1 (F_1,86_ = 16.4, *p*-value < 0.001). The OIs then increased significantly in both areas 2 months later, after which they remained relatively stable within each site type (Figure 3). However, the OI increased significantly more in the control areas than in the intervention areas at Month 2 (F_1,85_ = 13.1, *p*-value < 0.001), remaining as such in Months 3 (F_1,85_ = 28.9, *p*-value < 0.001) and 4 (F_1,84_ = 5.85, *p*-value = 0.018).

Over the months following intervention implementation, there were 22, 8, and 3 seroconversions at Months 2, 3, and 4 in the intervention area, respectively, and 2, 6, and 7 in the control area for the same months of data collection (Table 2). No information was available concerning their symptomatic status potentially associated with the presumed infection. In Month 2, 16 clusters registered seroconversions in the intervention sites; 1 with 4 seropositives, 3 with 2 seropositives, and 12 with 1 seropositive. In the 21 houses harboring a seroconversion in the intervention area, only one of the six houses containing more than one tested individual yielded a second seroconversion. In the control site, there were two single seropositive clusters. Of the two houses harboring a seroconversion, one contained more than one person tested, but only one seroconversion was observed. In Month 3, there were two clusters with two seropositives and four clusters with single seropositives at the intervention site. These seroconversions occurred in seven houses; three of these houses had more than one person sampled and one yielded two seroconversions. In the control site, there was one cluster with two seroconversions and four with single seropositives, with one seroconversion occurring in each of the six houses. Only one of these houses had more than one person sampled. In Month 4, there were three single seroconversion clusters at the intervention site, occurring in three different houses. There were seven single seroconversion clusters at the control site. Five of the houses with a seroconversion contained more than one sampled person, but none yielded more than one seroconversion.

In a global statistical analysis across all months, there was no association of seroconversion with any of the demographic and mosquito data of the same month (Arcsin%OI: χ^2^_1_ = 1.89, *p*-value = 0.17; Arcsin%CI: χ^2^_1_ = 0.45, *p*-value = 0.502; Age χ^2^_2_ = 0.47, *p*-value = 0.47; Sex χ^2^_1_ = 1.55, *p*-value = 0.214). Neither Arcsin%OI nor sex was found to be associated with seroconversion in the multivariate analysis. Lagged mosquito indices were also analyzed for any association with subsequent seroconversion rates, but none were found to be significant.

As might be expected from a large number of seroconversions in the intervention site in Month 2 (Table 2) and a subsequent decrease in the following months, there was a significant interactive effect between site type and month (Sitetype.month: χ^2^_2_ = 20.02, *p*-value < 0.001). Following the higher OR of seroconversion in Month 2 in the intervention site (OR = 12.16 95%CI 3.64–40.59), the ORs decreased steadily in Months 3 and 4 in the intervention site (Month 3: OR = 1.51 95%CI 0.57–3.98; Month 4: OR = 0.46 95%CI 0.15–1.43). This sharp decrease from Month 2 contrasts with the control site where the ORs increased substantially in Months 3 and 4 (Figure 4). As above, no variables were found to be associated with IgG ODs. However, a similar trend was observed when examining the changes in IgG optical densities over time (Figure 5). Intervention was not, however, observed to be significantly associated with a decrease in seroconversion or IgG optical density, even at Month 4 of intervention (Month 4 Intervention vs. Control: Seroconversion t = 1.34, *p*-value = 0.09; IgG Optical density t = 1.17, *p*-value = 0.122).

## 4. Discussion

Numerous studies have assessed the entomological efficacy of the application of PPF, whether through its application in water sources or by using an auto-dissemination approach where the mosquito would transfer small particles of PPF to other oviposition sites [28,29,30,31,32,33,34,35]. With the application of a granular formulation of PPF to ovitraps in study houses, we observed a decrease in ovitraps and the container index after two months of intervention. The more rapid reduction in the ovitrap index in the intervention sites, as compared to the container index, likely reflects the fact that the former measures egg presence whilst the latter measures larval presence, likely produced from enclosed eggs laid during the first month when the impact upon the adult mosquito population density was not yet affected. The low density of treated ovitraps (one per house) might not be expected to generate such a reduction on its own, impacting only on the eggs laid in the ovitraps. However, auto-dissemination of insecticide-contaminated water has been shown to occur, albeit at a much lower efficacy than the transfer of the powder form of PPF [36,37,38]. Additionally, females exposed to such contaminated water bodies have subsequently been observed to have reduced fertility [39]. The combination of these insecticidal effects may have contributed to the decrease in mosquito indices observed. It is notable that nearly 50% of the study clusters did not have any water-filled containers, thus limiting the number of available and competing artificial oviposition sites. The efficacy of larvicidal traps in a community setting to reduce mosquito density, which became the basis of the integrated vector control system of Metro Manila, has previously been observed when using Novularon [40]. Because of the relatively low flight distance of *Aedes aegypti*, especially in urban settings, the CI will reflect the mosquito population in the area immediately surrounding the household.

Our study is the first in the Philippines to use the immunological method for dengue diagnosis by measuring IgG from collected saliva samples. However, the measurement of IgG response levels among the participants is not very specific, cross-reacting with other flaviviruses, and it is not helpful in identifying dengue serotypes [41]. In our setting, there were no case reports of co-circulating flaviviruses. The use of saliva as an alternative to sera to detect levels of antibodies, including IgG levels, has been reported to be effective in dengue and other diseases [24,39,42,43,44]. A study demonstrated the efficacy of using salivary IgG levels to determine primary- and secondary-DENV infections, with high sensitivity and specificity for detecting anti-DENV IgG levels [39]. Future studies on the use of saliva to diagnose dengue infection may be an alternative to serum collection, which may be warranted when the collection of specimens is hindered by several factors, such as the age of patients or health facilities.

The current implementation of DOH strategies on the control of mosquito densities may be complemented with the use of pyriproxyfen-treated ovitraps with some modifications. However, implementing pyriproxyfen-treated ovitraps in the community should be guided by standard operating procedures and be closely monitored to ensure accuracy and consistency. Numerous considerations need to be addressed in order to proceed with pyriproxyfen-treated ovitraps implementation, including distribution schemes, ovitrap maintenance, monitoring and evaluation procedures, and plans for disposing of used and unused ovitraps. Whether or not local communities will accept the implementation of these new strategies should also be evaluated and considered to ensure the cooperation of the public [36]. However, despite these issues, which need to be considered for any future upscaling and implementation of pyriproxyfen-treated ovitraps, using a small community in Taguig City, our study provided insights on its efficacy which may be used as evidence in further studies in the Philippines.

This study has several limitations, not least of which is the small human sample size. Such a sample size would require a powerful impact of PPF treatment to be detectable. Although the observed decrease in the seroconversion rate was not significant, the trend was clear and encouraging for rolling out a larger-scale study. Secondly, local structural urban topography can significantly affect adult mosquito flight range, impeding PPF dispersal capacity and/or exposure to contaminated water bodies [45,46]. In addition, our study did not consider the heterogenicity of characteristics found in urban areas. This heterogeneity should be optimized in future research designs and be taken into account through fine-scale spatial analyses. Moreover, our study is limited to the specific doses of PPF deployed using a particular design of ovitraps. Different designs of ovitraps and dosing of PPF will undoubtedly impact efficacy [46,47]. Finally, our study only collected data during the cool-dry season. Future studies on the effect of PPF treatment in reducing dengue infection may differ during the dry season compared to the rainy season when oviposition sites are more numerous and diverse. More extensive studies on the effectiveness of PPF, which will monitor medium to long-term effects, are warranted to confirm our findings.

## 5. Conclusions

A total of 443 participants were included in this study, of which 220 belonged to the intervention group and 223 belonged to the control group. As a baseline, 17 and 13 of the participants were seropositive for dengue in the intervention and control groups, respectively. In the intervention group, entomological indices were lower compared with the control area at Months 2, 3, and 4 post-interventions, but not earlier. There was also an observed reduction in dengue seroconversion rates, but a not significant one. Therefore, the use of PPF-treated ovitraps may have impacted the mosquito population but not the seroconversion rates. Continuing evidence regarding the entomological efficacy of PPF leaves us optimistic that dengue incidence might be reduced if the treatment is implemented strategically. This may complement the existing vector control strategies that the local health authorities are currently implementing. Moreover, our results demonstrated a need to extend the exposure to PPF and to saturate all houses in a community in order to present a significant reduction in dengue incidence.

## Figures and Tables

**Figure 1 ijerph-19-03026-f001:**
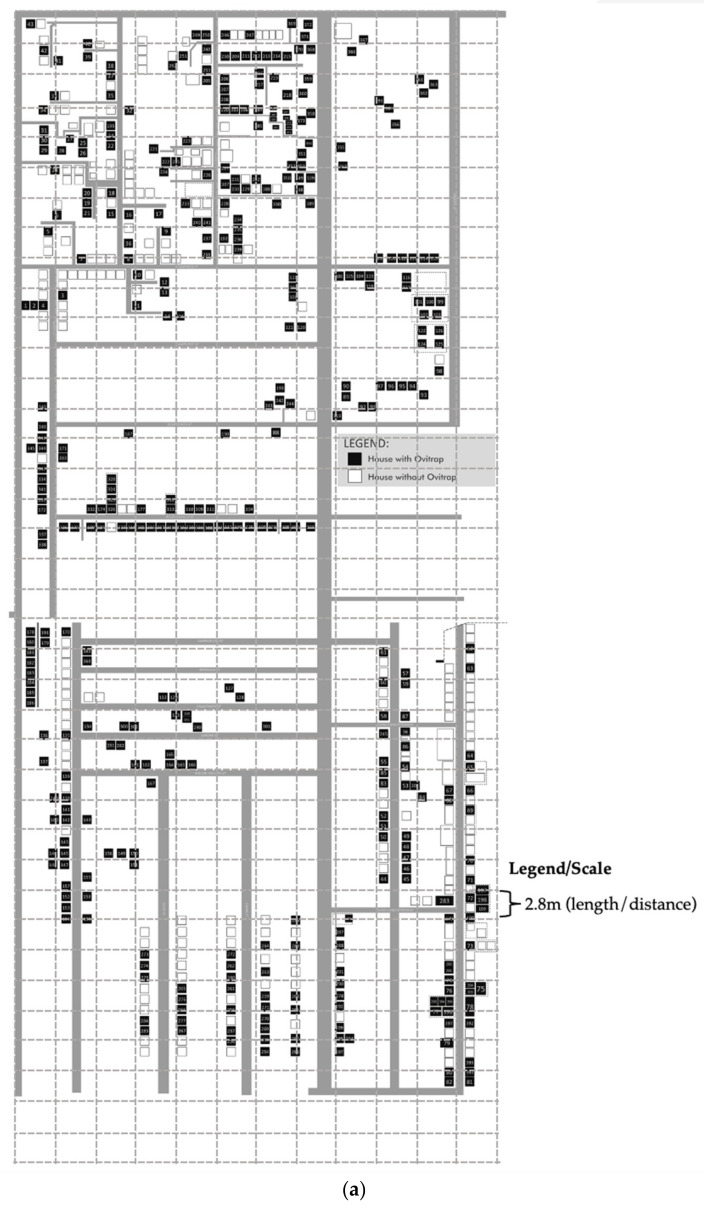
(**a**) Schematic Trial Design for the Control Area (Katipunan Area). (**b**) Schematic Trial Design for the Intervention Area (PNR Area).

**Figure 2 ijerph-19-03026-f002:**
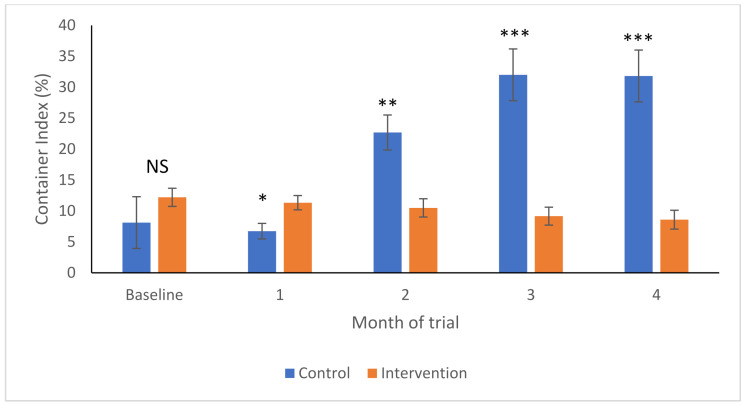
Container indices (Arcsine transformed to yield mean and SEM) in intervention (orange) and control (blue) areas over time. NS—no significant differences; * *p* < 0.05; ** *p* < 0.01; *** *p* < 0.001.

**Figure 3 ijerph-19-03026-f003:**
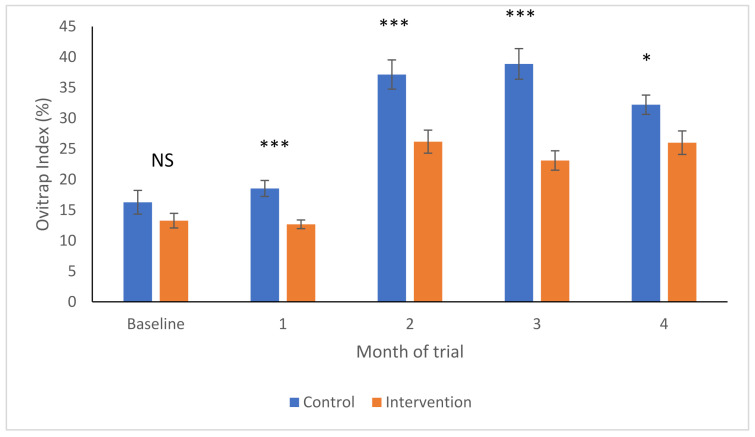
Ovitrap indices (Arcsine transformed to yield mean and SEM) in intervention (orange) and control (blue) areas over time. NS—no significant differences; * *p* < 0.05; *** *p* < 0.001.

**Figure 4 ijerph-19-03026-f004:**
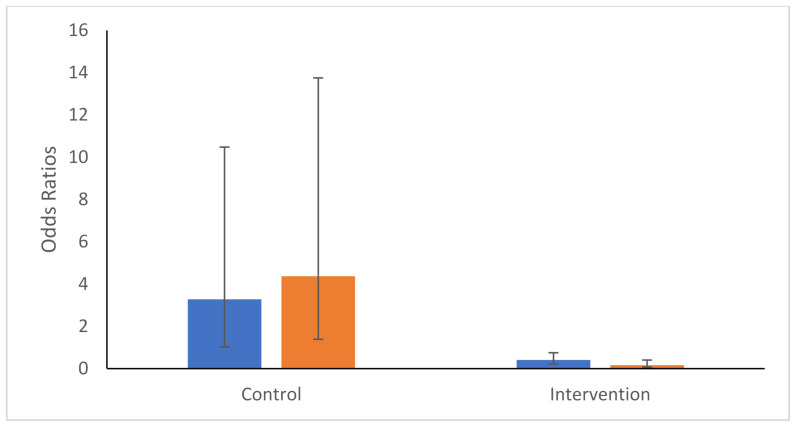
Odds Ratios and 95% confidence intervals of seroconversion probabilities for Months 3 (blue) and 4 (orange) compared with Month 2′s rates within each intervention type (control or intervention).

**Figure 5 ijerph-19-03026-f005:**
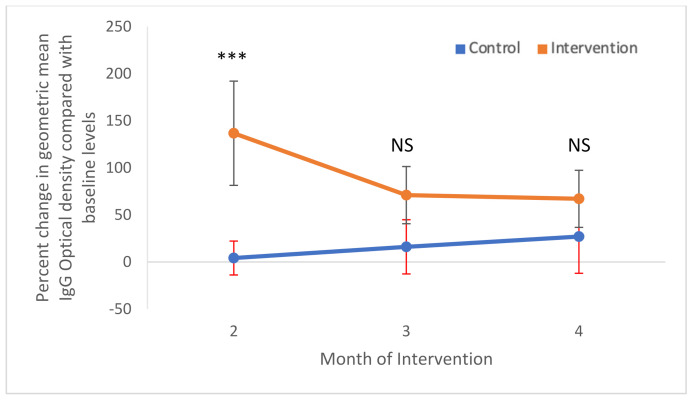
Percent change in geometric mean IgG optical densities as compared with the baseline. Shown are means and 95% confidence intervals for control (blue) and intervention (orange) areas. NS—no significant differences; *** *p* < 0.001.

**Table 1 ijerph-19-03026-t001:** Baseline characteristics of the study population, by the site (*n* = 443).

Characteristics	Intervention Group		Control Group		Total	
	*n*	%	*n*	%	*n*	%
Age, years old						
Child, 1–12	99	45	73	32.8	172	38.83
Adolescent, 13–18	60	27.3	38	17	98	22.12
Young adult, 19–30	61	27.7	112	50.2	173	39.05
Sex						
Male	103	46.8	103	46.2	206	46.50
Female	117	53.2	120	53.8	237	53.50
Income, PhP						
No income	3	1.4	19	8.5	22	4.97
1 to <10,000	41	18.6	48	21.5	89	20.09
10,001 to <20,000	161	73.2	123	55.2	284	64.11
20,001 to <30,000	14	6.4	24	10.8	38	8.58
30,001 and above	1	0.4	9	4	10	2.26
Mother’s Education						
At least High School	199	90.5	168	75.3	367	82.84
College/Postgraduate	20	9.1	51	22.9	71	16.03
Deceased/No Data	1	0.4	4	1.8	5	1.13
Father’s Education						
At Least High School	183	83.2	164	73.5	347	78.33
College/Postgraduate	26	11.8	57	25.6	83	18.74
Deceased/No Data	11	5	2	0.9	13	2.93
Average IgG, g/L	220	0.046 ± 0.036	223	0.044 ± 0.033		

**Table 2 ijerph-19-03026-t002:** Distribution of dengue seropositive individuals during the study.

Groups	Number of Dengue Seropositive Individuals			
	Baseline	Month 2	Month 3	Month 4
	Intervention (*n* = 220)	Intervention (*n* = 203)	Intervention (*n* = 181)	Intervention (*n* = 173)
	Control (*n* = 223)	Control (*n* = 210)	Control (*n* = 208)	Control (*n* = 202)
Intervention	17	22	8	3
Control	13	2	6	7

## Data Availability

The authors confirm that some access restrictions apply to the data. The researchers interested in using the data must obtain approval from the St. Luke’s Medical Center’s College of Medicine Research Ethics Committee. The researchers using the data are required to follow the terms of a number of clauses designed to ensure the protection of privacy and compliance with the relevant Data Privacy Act of the Philippines. Data requests may be subject to further review by the Ethics Committee and may also be subject to individual participant consent.

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
