# Peer review of "Assessing Entomological and Epidemiological Efficacy of Pyriproxyfen-Treated Ovitraps in the Reduction of Aedes Species: A Quasi-Experiment on Dengue Infection Using Saliva Samples"

_ijerph, 2022, doi:10.3390/ijerph19053026_

Round 1

Reviewer 1 Report

Ligsay et al submitted a very interesting manuscript describing the effect of PPF-treated ovitraps on numbers of mosquitoes and new DENV seroconversion in a wide metropolitan area of Manila. Despite the fact that they showed that use of the PPF-treated ovitraps led to a decrease in efficiency of mosquitoes reproduction, they did not observe any significant effect on DENV transmission. This observation can have numerous reasons, which are listed in the Discussion. I would add that some effect could be observed if the experiment would not be terminated after four months but continued for a longer period if possible (e.g., due to the climatic conditions). I want to ask the authors to discuss this possibility in their manuscript. I also appreciate the use of saliva as an easy-to-get noninvasively collected sample for antibody detection.

The manuscript is written in a clear, easy-to-understand, and straightforward style. I have only several minor comments which I would like to ask the authors to clarify:

Methods:

Figure 1a – According to the legend in Figure 1a the houses without ovitrap should be shown as white squares. There is only one white square on the map, and the remaining squares are grey.

Line 114 – Why was the study limited only to individuals 1-30 years old? Is it because you expected lower seropositivity in this age group?

Line 167 – It would be great to state at least an approximate ratio between PHP and USD/EUR.

Results:

Line 217 – Is it possible that a significant difference in income between the control and intervention groups could have some impact on the observed results? I would suggest discussing this difference and its possible impact in the Discussion.

Discussion:

Line 320 – What serotypes of DENV were recently observed in the study area?

Line 322 – References 37-39 describe the potential use of saliva for Ig detection in case of leptospirosis, HIV, and measle but not DENV as mentioned in the text. Another reference (e.g., 24) should be added here.

Conclusions:

I do not think that the statements in conclusions correspond to the results of the manuscript. I would suggest rewriting the conclusions completely and focusing only on the results of the research.

Reviewer 2 Report

The MS by Ligsay et al present results on a field study performed to test the efficacy of traps for A. aegypti containing a classic insecticide, PPF, this monitored by assessing the population seroconversion using saliva as sample. Though the tackled health problem is a real issue worldwide and particularly in the studied region, the MS has several flaws and the obtained results do not add to the current literature. This because the MS present 2 main results: (1) the efficacy of PPF-containing traps for controling mosquitos infestation levels. This result is not a novelty, as it is already mentioned in the Introduction section, where the Authors should also ackowledge the numerous existing reports on the use of this kind of traps, e.g. Marina et al 218 Parasitology Res; Ponlawat et al 2013 Southeast Asian J Trop Med Public Health. (2) the monitoring of seroconversion using saliva. The approach is not new, as the Authors mentioned. Additionaly, the MS does not present any significant result from the use of this approach to answer any question. The MS actually shows that there were more seroconversion in the treated study population. Thus, I can not be positive in recomending this MS for publication in ijerph. The MS might be of interest for the local communities so that it can be submitted to a public health jornal of the country where the study was performed. There follow some details: 

Title:

“Assessing entomological and epidemiological efficacy of pyriproxyfen-treated ovitraps using saliva samples” Should include a suggestion on what kind of insect or disease.

  Introduction

Line 37, Ref. #1: Please, consult an updated version, since this is 10 years old. Line 68, Ref. #27: Poisson regression analysis. This ref. does not look appropriate here.

Please, cite appropriately all WHO references, eg. For example, ref. #22:

World Health Organization & UNICEF/UNDP/World Bank/WHO Special Programme for Research and Training in Tropical Diseases. (‎2017)‎. Global vector control response 2017-2030. World Health Organization. https://apps.who.int/iris/handle/10665/259205. Licença: CC BY-NC-SA 3.0 IGO

 Line 82, Ref. #32: there is a problem with this reference as well. Please, double-check all references. Lines 83-85: a clear elaboration on which approaches were tested and the aims of the study should be presente here.  

M&M

Maps (Figure 1) should include some kind of scale. Line 114: 1-30 years old? Is there any concern or particularity on the immune response of such a broad spectrum of targeted age group, especially at those early ones? Somewhere the MS should mention when the Project (field work) was performed (year, months of the year, etc.). The MS should describe all the available epidemiological data on dengue. 

Thoroughly describe Container index and Ovitrap index measures, the respective devices and the rationale of their use. The present description does not seem to show diferences between both types of devices. This will help the reader to get conclusions from data presented in figures 2 and 3.

  Results

The results should be grouped and presented in separate sections using informative titles. This will make easier for the reader to get the information. Figure and table legends should be self-explaining and should provide easy and sufficient information without referring to the main text [eg, in figure 2: “treatment (Orange)”, briefly mention the kind of treatment]. In addition, all graphs should show marks on the bars representing significantly different means, which should be described in the respective caption as well. All this will broader the audience of the paper, what should be one of the main aims of this type of study. Line 252-268: all this info should be in a table. It is too hard to follow only in the present format. The same with all data. Info of figure 4 should be presented in a table. 

Discussion

The MS should present an elaboration on the difference in treatment results as measured by containers and ovitraps (see suggestion on M&M). 

Line 314: it is A. aegypti.

 Lines 317-319: include a reference for this affirmation. Lines 349-350: this idea is hard to evaluate since the MS does not have enough data on the experimental setting used in the study (see above). Lines 351-352: this information is also lacking in the M&M section. 

Lines 357-361: these are not actually conclusions. This because the MS fails in presenting real novelties, in spite of the hard work of spreading the traps and taking baseline data on the study population. 

Overall, the MS would benefit from punctuation and spelling editing (eg. Section 2.3.1. title).

Reviewer 3 Report

Only, I suggest changing in region 316 the term serological for immunological, since saliva is named a component of blood, the presence of antibodies in it is due to the presence of secretory antibodies produced by the plasma cells of the mucous membranes. I also recommend eliminating the graphs that do not provide useful information, and on the contrary are confusing
